# Evaluation of physical activity calorie equivalent (PACE) labels' impact on energy purchased in cafeterias: A stepped-wedge randomised controlled trial

**James P. Reynolds**[1,2]\*, **Minna Ventsel**[1], **Alice Hobson**[1,3], **Mark A. Pilling**[1], **Rachel Pechey**[3], **Susan A. Jebb**[3], **Gareth J. Hollands**[1,4], **Theresa M. Marteau**[1]\*

1 Behaviour and Health Research Unit, University of Cambridge, Cambridge, United Kingdom, 2 School of Psychology, Aston University, Birmingham, United Kingdom, 3 Nuffield Department of Primary Care Health Sciences, University of Oxford, Oxford, United Kingdom, 4 EPPI-Centre, UCL Social Research Institute, University College London, London, United Kingdom

\* j.reynolds4@aston.ac.uk (JPR); tm388@medschl.cam.ac.uk (TMM)

## Abstract

### Background

A recent meta-analysis suggested that using physical activity calorie equivalent (PACE) labels results in people selecting and consuming less energy. However, the meta-analysis included only 1 study in a naturalistic setting, conducted in 4 convenience stores. We therefore aimed to estimate the effect of PACE labels on energy purchased in worksite cafeterias in the context of a randomised study design.

### Methods and findings

A stepped-wedge randomised controlled trial (RCT) was conducted to investigate the effect of PACE labels (which include kcal content and minutes of walking required to expend the energy content of the labelled food) on energy purchased. The setting was 10 worksite cafeterias in England, which were randomised to the order in which they introduced PACE labels on selected food and drinks following a baseline period. There were approximately 19,000 workers employed at the sites, 72% male, with an average age of 40. The study ran for 12 weeks (06 April 2021 to 28 June 2021) with over 250,000 transactions recorded on electronic tills. The primary outcome was total energy (kcal) purchased from intervention items per day. The secondary outcomes were: energy purchased from non-intervention items per day, total energy purchased per day, and revenue. Regression models showed no evidence of an overall effect on energy purchased from intervention items, −1,934 kcals per site per day (95% CI −5,131 to 1,262), *p* = 0.236, during the intervention relative to baseline, equivalent to −5 kcals per transaction (95% CI −14 to 4). There was also no evidence for an effect on energy purchased from non-intervention items, −5 kcals per site per day (95% CI −513 to 504), *p* = 0.986, equivalent to 0 kcals per transaction (95% CI −1 to 1), and no clear evidence for total energy purchased −2,899 kcals per site (95% CI −5,810 to 11), *p* = 0.051, equivalent

**Data Availability Statement:** This study includes sales data from the collaborating catering company, which were received weekly from April

2021 to June 2021. Due to contractual restrictions to the use of the sales data, these data cannot be made openly available. Any requests to access the data can be directed to our administrators (bcbd. administrator@medschl.cam.ac.uk). The collaborating catering company wished to remain anonymous and any requests for further use of these data must be sent via our administrators who will send the request to the collaborating company.

**Funding:** This report is independent research funded by a collaborative Award in Science from Wellcome Trust (Behaviour Change by Design: 206853/Z/17/Z) awarded to Theresa Marteau, Paul Fletcher, Gareth Hollands, and Marcus Munafò. Rachel Pechey is supported by a Wellcome Trust Research Fellowship in Society and Ethics [106679/Z/14/Z]. The University of Cambridge was the study sponsor. The funders and sponsor were not involved in the study design, data collection, management, analysis, interpretation, writing, or submission of this manuscript.

**Competing interests:** The authors have declared that no competing interests exist.

to −8 kcals per transaction (95% CI −16 to 0). Study limitations include using energy purchased and not energy consumed as the primary outcome and access only to transaction-level sales, rather than individual-level data.

## Conclusion

Overall, the evidence was consistent with PACE labels not changing energy purchased in worksite cafeterias. There was considerable variation in effects between cafeterias, suggesting important unmeasured moderators.

## Trial registration

The study was prospectively registered on ISRCTN (date: 30.03.21; ISRCTN31315776).

---

## Author summary

### Why was this study done?

- Overconsumption of energy from food and drink contribute to high and rising rates of obesity.
- PACE (physical activity calorie equivalent) labels have been suggested as one intervention that could lower consumption of high energy foods.

### What did the researchers do and find?

- We carried out a randomised stepped-wedge trial over 12 weeks.
- PACE labels were added to a range of food and drinks in 10 cafeterias after a baseline period.
- The labels did not significantly change overall energy purchased in the cafeterias, although there was considerable variation in the effects between cafeterias.

### What do these findings mean?

- There was no evidence of an overall effect of PACE labels on energy purchased in cafeterias in this study.
- There may be important contextual factors that influence when PACE labels do and do not reduce energy purchased.

## Introduction

Excess energy intake contributes to over 60% of the United Kingdom adult population being overweight or obese, contributing to high and rising levels of type 2 diabetes and 13 different types of cancer [1–3]. Interventions to reduce overconsumption of energy must form a central part of wider strategies to tackle overweight and obesity.

One key setting in which to intervene is the out-of-home setting including eating establishments such as restaurants, cafes, and schools. This setting is important to target as, in the UK, up to 33% of meals are eaten out-of-home [4], and the energy content of these products is often excessive [5], with 1 study estimating out-of-home meals to be around 31% more calorie dense than those eaten in the home [6]. One approach to reducing excess energy intake has been to add labels on food and drinks to inform people about the energy content of the product. A Cochrane systematic review and meta-analysis of 3 nutritional labelling studies in restaurants suggested a reduction in energy purchased by 47 kcals per meal [7], whereas a separate meta-analysis on 6 labelling studies in restaurants concluded that there was no effect on energy ordered [8]. However, the quantity and quality of the available evidence is limited. Two randomised trials in worksite cafeterias published after these reviews provided no evidence for an effect of simple energy labelling (kcal) on energy purchased [9,10].

An alternative to labelling the energy content is to convert this information into the physical activity needed to expend the energy in that product. Physical activity calorie equivalent (PACE) labels typically include the energy content, the equivalent energy in terms of physical activity, and an image representing the type of physical activity—usually walking or running. These labels aim to help consumers understand the energy content of the products and thus reduce excess energy intake [11]. A recent systematic review concluded that PACE labels may reduce energy selected from menus and decrease the energy consumed when compared to no labelling or other types of labelling such as kcal labelling [11]. However, of the 15 included studies, most were of an unclear risk of bias and only 1 was conducted in a naturalistic setting [12]. The remaining 14 studies were conducted online ($n = 8$) or in non-naturalistic settings ($n = 6$), and recent reviews of labelling studies suggest that effects are typically largest in online studies and smallest in naturalistic settings [8,13]. One naturalistic study [12] investigated the effect of PACE labels on sugar-sweetened beverages in 4 convenience shops in the United States of America. The results suggested that participants were less likely to purchase a sugar-sweetened beverage when PACE labelling was added (OR = 0.51). A further naturalistic study in 3 worksite cafeterias, published after the review, found that PACE labels resulted in a significant decrease in energy purchased of approximately 40 kcals per meal [14]. This study recruited participants who regularly used the cafeterias and asked them to photograph their food at multiple points over time. This study therefore only includes a subset of the total customers and a subset of the total purchases that these participants made over the study period. Based on the quantity and quality of evidence, considerable uncertainty remains about the effect of PACE labels to reduce energy purchased and consumed. Furthermore, there is evidence that effects reported in online or lab studies will be significantly smaller or even nonexistent when implemented in naturalistic settings [8,13]. Even if an effect is replicated in a real-world setting, the high variability in contexts make it hard to predict if an effect reported in 1 eating retail outlet will generalise to another.

The aim of the current study is to estimate the effect of PACE labels on energy purchased in worksite cafeterias. The limitations of existing naturalistic studies are addressed in 3 ways: implementing the intervention in cafeteria settings across a wide range of food and drinks; collecting sales data from electronic tills to ensure the availability of data on every purchase made by every customer of the cafeterias throughout the study period; and conducting the study in a

larger number of sites to increase the study power and test the generalisability of the main effects to multiple cafeterias. The study is designed to test the hypothesis that customers purchase less energy when food and drinks feature PACE labels.

## Methods

The study was prospectively registered on ISRCTN (ISRCTN31315776) and a detailed analysis plan was uploaded to the Open Science Framework (https://osf.io/2a5cg/?view_only=95d3d6a38cf047588f4e8365207ef1f4) during data collection, but before data cleaning or analysis had commenced. There was 1 main deviation from the prespecified analysis plan: We did not conduct the second prespecified sensitivity analysis (see Analysis section for details). The study protocol and CONSORT extension checklist for stepped-wedge trials are attached as supplements (see "S1_CONSORT_Checklist" and "S1_Study_Protocol"). The Cambridge Psychology Research Ethics Committee based at the University of Cambridge approved the trial on 08.12.20 (No. PRE.2020.105). The research team obtained informed and written consent from a representative of the catering company on behalf of the participating cafeterias.

### Cafeterias

Ten worksite cafeterias were recruited through a major UK catering company (see Fig 1) and were based within worksites belonging to different companies. There were 4 eligibility criteria for participation: (i) at least 500 employees based at the cafeteria; (ii) sales data recorded using electronic point-of-sale tills; (iii) able to provide kcal information for all food and drink sold; and (iv) an absence of existing calorie labels. Twenty cafeterias were screened for eligibility and 10 participated in the study (Fig 1). The remaining 10 cafeterias were not eligible due to violating the first eligibility criterion. At recruitment, participating sites employed between 500 and 7,200 staff. The smallest cafeteria, Cafeteria 8 (230 employees), had fewer employees by the end of data collection than was reported during recruitment due to COVID-related staffing changes (see Table 1). For further information on the products sold at each site, see Table A in S1 Additional Data in the Supporting information.

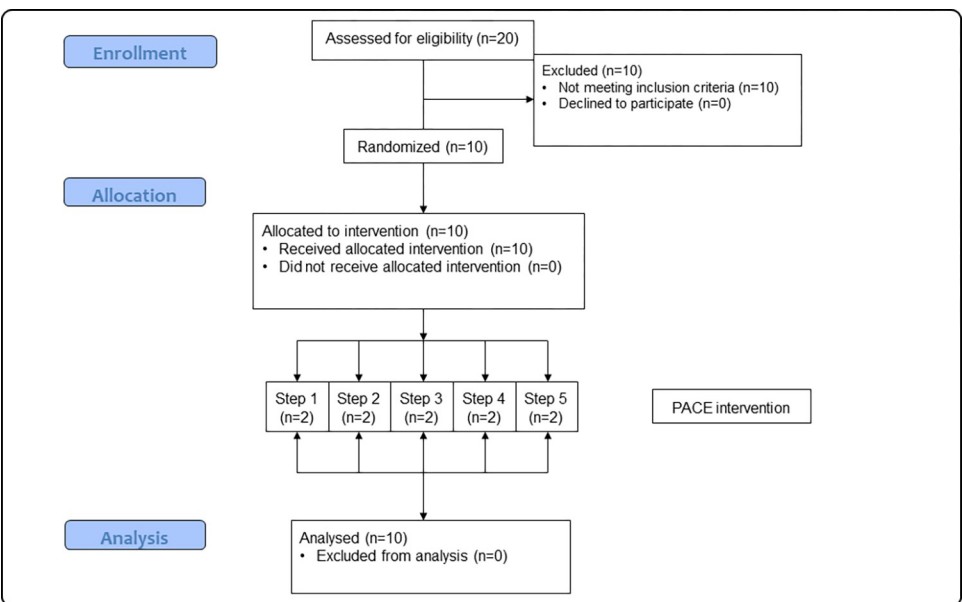

**Fig 1. Cafeteria inclusion flowchart.** *Note. n* refers to cafeterias. Each step refers to the different weekly periods in which cafeterias start the interventions.

**Table 1. Demographic characteristics of employees in participating sites.**

| Cafeterias | Number of employees | Male (%) | Age (mean) | Typical job roles | Full time (%) |
|---|---|---|---|---|---|
| Total | 19361 | 72 | 40 | - | 77 |
| 1 | 1800 | 70 | 37 | Pilots/ Office/ Admin/ Facilities/ Management | 80 |
| 2 | 697 | 75 | 47 | Processing/ Drivers Office Staff | 62 |
| 3 | 1500 | 80 | 44 | Engineers/ Designers/ Accountants/ Facilities/ Management | 100 |
| 4 | 2361 | 80 | 26 | - | - |
| 5 | 850 | 70 | 45 | Processing/ Drivers Office Staff | 60 |
| 6 | 3473 | 95 | 39 | Manufacturing associates/Admin / Management | 97 |
| 7 | 500 | 60 | 47 | Processing/ Drivers Office Staff | - |
| 8 | 230 | 50 | 30 | Manufacturing/ Management & Office | 100 |
| 9 | 750 | 66 | 42 | Drivers/ Office/ Facilities/ Sales/ Admin/ Delivery/ Managers | 40 |
| 10 | 7200 | - | - | - | - |

*Note*: Some data are missing as the cafeterias managers did not provide it.

The recruitment strategy was based on practical limitations, specifically, the maximum number of eligible cafeterias that we could recruit from our collaborating company. We aimed to recruit 10 cafeterias, an increase compared to similar calorie labelling studies in cafeterias [9,10]. An illustrative power calculation suggests that this would provide 80% power to detect a change in energy purchase of Cohen's $d = 1.00$, using a before-after repeated measures design with 4 weeks in each period, using a 2-sided test and at the 5% significance level using a paired $t$ test. The stepped-wedge design (see Fig 2) was used for pragmatic reasons, as they are typically preferred to a parallel groups randomised controlled trial (RCT) when study resources only allow a staggered implementation of the intervention(s) [15].

## Study periods

**Baseline.** Baseline was a period of business-as-usual for the cafeterias when sales data were collected before the intervention period. Photos were checked (see Fidelity section below) to confirm that during the baseline period, no PACE labels or other prominent energy labels were in use. During the baseline period, most preexisting labels and menus only featured the product name and price. There were some standardised front-of-pack nutrition labels on branded products (e.g., Coca-Cola) and in-house products (e.g., muffins) on which the energy content was provided in small print along with other nutritional information. There was no energy information on shelf-edge labels or menus beyond this standardised nutritional information.

**Intervention.** The PACE label intervention comprised adding 2 new pieces of information to the product: the energy content of the product and the PACE of this value, expressed in the minutes of walking that would be needed to expend the energy in the product (see Fig 3). Previous studies have evaluated different variants, for example, running versus walking and minutes versus miles. In the absence of any evidence of their relative effectiveness, the investigators selected the design they judged to be the most accessible and easily understood, namely communicating the number of minutes the average person would need to spend walking to expend the energy contained in the product. Initial versions were developed by the research team based on designs reported in previous published papers. Feedback was then obtained from the catering staff and managers working at the study sites. After several iterations, we agreed on the selected design. In the typology of interventions in proximal physical microenvironments (TIPPME) [16], this is classified as an *Information x Product* intervention. These

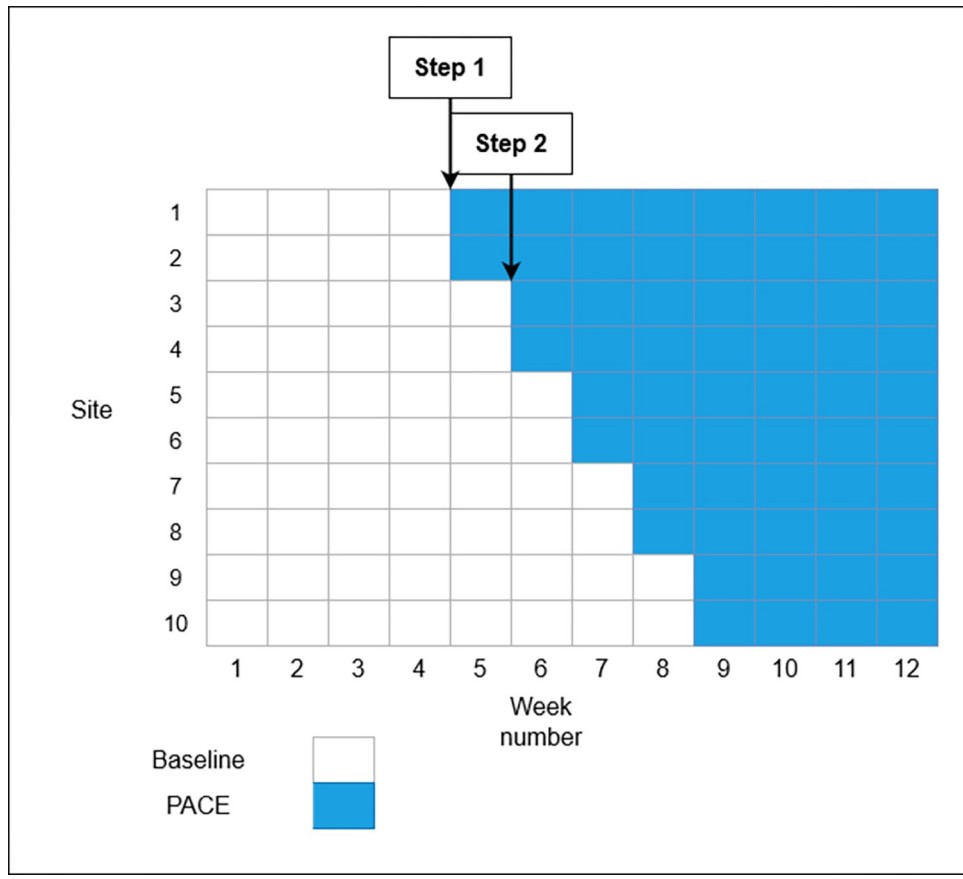

**Fig 2. Stepped-wedge study design.**

labels were added in up to 4 locations at each cafeteria: (i) shelf-edge labels; (ii) menus next to food and drink displays; (iii) individual tent cards next to food and drink displays; and (iv) on stickers that were attached to the product packaging. Posters that explained the meaning of the labels were also put up at participating cafeterias, and the service staffs were briefed in case customers asked them any questions.

## Study design

A stepped-wedge design was used across a period of 12 weeks (06 April 2021 to 28 June 2021). The 10 cafeterias were randomly allocated to the week in which the intervention was implemented (see Fig 2). The baseline period lasted between 4 and 8 weeks. Weeks 1 to 4 comprised the minimum baseline period. From week 5 until week 9, 2 cafeterias a week implemented the PACE label intervention, which lasted until the end of week 12, when the study ended.

## Randomisation and blinding

Participating cafeterias were randomly allocated to the time at which the interventions were implemented. The randomisation was performed by a Statistician who was blinded to the identity of the cafeterias. The Statistician allocated a list of anonymised cafeteria names using the rank of random numbers from Excel. Staffs at each cafeteria were told not to inform customers that the labels were part of research. In response to questions from customers about the labels, staffs were instructed to say they were being trialled as part of a health initiative.

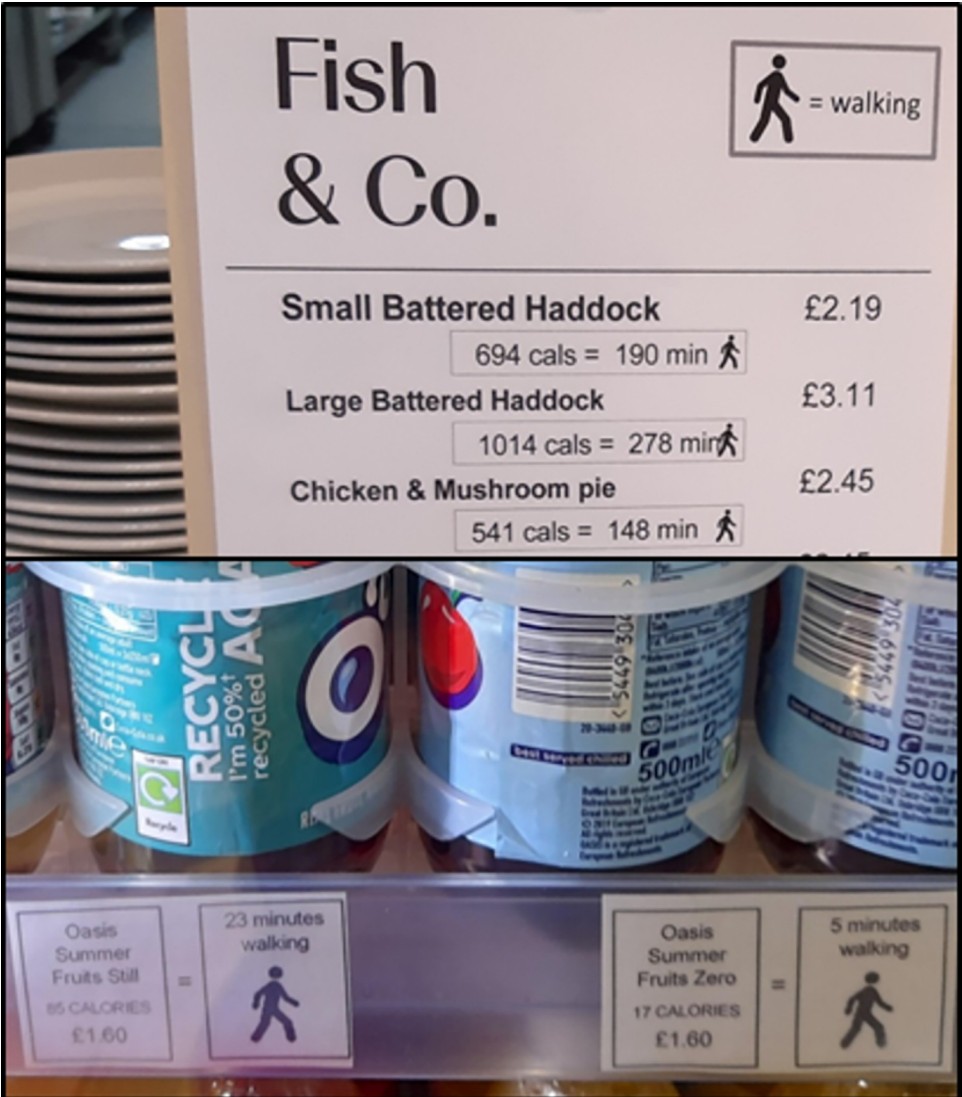

**Fig 3. Example menu (top) and shelf-edge labels (below) used during the PACE intervention.**

Staffs and customers could not be blinded to the interventions as the labels are designed to be seen and read by customers. Cafeterias were informed about their allocation (the week in which they were to implement the interventions) after recruitment and before data collection, which allowed time for them to prepare for the interventions.

### Procedure

**Implementation.**   The PACE labelling intervention was implemented by catering staffs and managers at each cafeteria following training and assistance from senior management within the catering companies and the research team. To maximise coverage, the catering staffs were instructed to add labels to all products on sale for which the energy content was known.

**Fidelity.**   Detailed photos of the products on sale and their labels were requested to be sent once a week to the research team for checks. During the baseline period, these checks were to

**Table 2. Implementation of PACE labelling overall and by cafeteria.**

| Cafeteria | Proportion (%) of total products that received PACE labels | Location: Shelf-edge labels | Location: Menus | Location: Tent cards | Location: Stickers on products |
|---|---|---|---|---|---|
| **Overall** | 93 | 10/10 | 9/10 | 5/10 | 1/10 |
| **1** | 97 | Yes | Yes | No | No |
| **2** | 100 | Yes | Yes | No | No |
| **3** | 92 | Yes | Yes | No | No |
| **4** | 93 | Yes | Yes | No | No |
| **5** | 95 | Yes | Yes | Yes | No |
| **6** | 96 | Yes | Yes | Yes | No |
| **7** | 90 | Yes | Yes | Yes | No |
| **8** | 92 | Yes | Yes | Yes | Yes |
| **9** | 84 | Yes | Yes | Yes | No |
| **10** | 90 | Yes | No | Yes | No |

ensure that (i) no PACE labels were present; and (ii) no energy (kcal) labels were present. During the PACE labelling period, these checks were to ensure that (i) the PACE labels were present; and (ii) the energy (kcal) and PACE values on these new labels were accurate.

These photos were used to calculate the proportion of products that received PACE labels as well as the locations that the labels were presented. A site was coded as presenting PACE labels in a location (menus, shelf-edges, tent card, on products) if at least 1 photo was provided which showed evidence of a label in that location (see Table 2).

Any violations to these criteria were reported to a manager responsible for the cafeteria with a request to rectify the violation and provide photographic evidence of the rectification within 24 hours. Adherence to or violation of the planned implementation was recorded for use in secondary analyses.

## Measures

All outcomes were calculated using data from the electronic point-of-sale tills that were used at each cafeteria. Data were collected during every day that the cafeteria was open during the length of the study.

**Primary outcome.** This comprised energy (kcal) purchased from intervention items per day. This was calculated using the total number of sales for all items that featured a PACE label and the energy content for each of these items.

To determine which products featured a PACE label, we used the digital copies of the PACE labels and menus. These digital copies were Word and PowerPoint documents that contained every PACE label and menu. These were printed off at each cafeteria and then added to the products on sale.

Energy content (kcals) was available for most products (97%) on sale at the cafeterias. This information was obtained from the catering provider, the cafeterias, and by searching online. For a further 16 products (1%), energy content was estimated by taking the average from 3 similar products, resulting in energy content for 98% of all products. For the remaining 2%, it was not possible to reliably estimate energy content using any approach, and therefore, these data were not included in the analysis.

**Secondary outcomes.**

1. Total energy (kcal) purchased per day from non-intervention items (products that did not include PACE labels).

2. Total energy (kcal) purchased from all food and drink products. This included all products, including intervention and non-intervention items.

3. Total revenue from each cafeteria. This was calculated from the total number of items sold in all categories and the price of each item.

**Covariates.**

1. Total number of transactions: the number of distinct payments to purchase products in the cafeteria, as a proxy measure for the number of customers per day.

2. Time: day number of the trial at each cafeteria, starting from 1 within Baseline or Intervention periods, common to all cafeterias.

3. Time (week of trial): common to all cafeterias, fitted as a random effect.

4. Time (day of the week).

## Analyses

Generalised additive linear mixed models [17] were used to estimate the overall potential impact of the PACE intervention compared to baseline due to markedly different variability (heteroscedasticity) at cafeterias. The number of transactions was included in the model as a proxy for site busyness. Cafeterias were fitted as random effects, with the effect of the day of the week allowed to vary by cafeteria as a random nested term due to regular weekly patterns at each cafeteria. The effect of week of the trial was fitted as a random factor common to all cafeterias, due to weekly changes observed in the data. To allow for any potential linear time trend, the day number of the trial from the period start (e.g., for site 1, this was valued as 1 to 25 in the baseline period and 1 to 56 in the intervention period) was fitted as a continuous fixed effect. Model results were later found to be stable when these time variables were excluded one-by-one from the final model. The final model was chosen based on minimising the Akaike information criterion. Additional ways of modelling time trends were considered, but the above approach was adopted as more complex models stopped model fitting or led to boundary singularities. Due to the irregular and rare instances of Cafeteria 3 opening at weekends, these 3 data points were removed as they were insufficient for parameters to be estimated.

When estimating the intervention effect at each of the 10 cafeterias, a Bonferroni adjustment for multiple testing was applied where the threshold for significance was 5%/10 and 99.5% confidence intervals were presented. Model diagnostics were assessed using variance inflation factors, residual plots, quantile–quantile plots, worm plots, and correlation function plots for the additive models; these diagnostics were acceptable. Exploratory plots of weekly-aggregate data were also examined, and heterogeneity was still present.

There were 2 prespecified sensitivity analyses. The first sensitivity analysis involves conducting a per-protocol analysis in which the data are re-analysed after accounting for intervention implementation. Cafeteria 10 failed to provide photographic evidence of label implementation on the assigned week (Week 9) and instead provided this during Week 12. This first sensitivity analysis removes this cafeteria's data from weeks 9 to 11, to account for the possibility that the labels were not implemented. The second prespecified sensitivity analysis was not conducted. This analysis planned to change the definition of "intervention items" from a product containing a label (e.g., crisps) to a product in which any product within its

category (e.g., savoury snacks) contained a label. However, as every product category contained at least 1 product with a label, this would have just replicated the second secondary outcome: total energy (kcal) purchased from all food and drink products.

A further sensitivity analysis was conducted. During data collection, researchers asked till staff at the cafeterias which if any buttons or barcodes were not working, meaning that certain products were mis-sold under another product's button or barcode. The primary analysis adjusts for this error by taking the average energy content for the multiple products that were sold under a specific till button, whereas the first sensitivity analysis reports the results without this adjustment.

## Results

### Implementation

All cafeterias reported implementing the labels on the planned week, with 9/10 cafeterias providing immediate photographic evidence of this implementation.

The fidelity checks suggested that the PACE labelling intervention was applied to 93% of products on sale (Table 2). The most common locations for the PACE labels to be added was on shelf-edges (10/10 cafeterias) and menus (9/10 cafeterias).

### Primary outcome

There was no evidence that the PACE intervention resulted in an overall change in energy purchased from intervention items, −1,934 kcals per site per day (95% CI −5,131 to 1,262), $p = 0.236$, equivalent to −5 kcals per transaction (95% CI −14 to 4). These results are shown in Fig 4, the full model is presented in Table 3, and the unadjusted data are shown in Table 4.

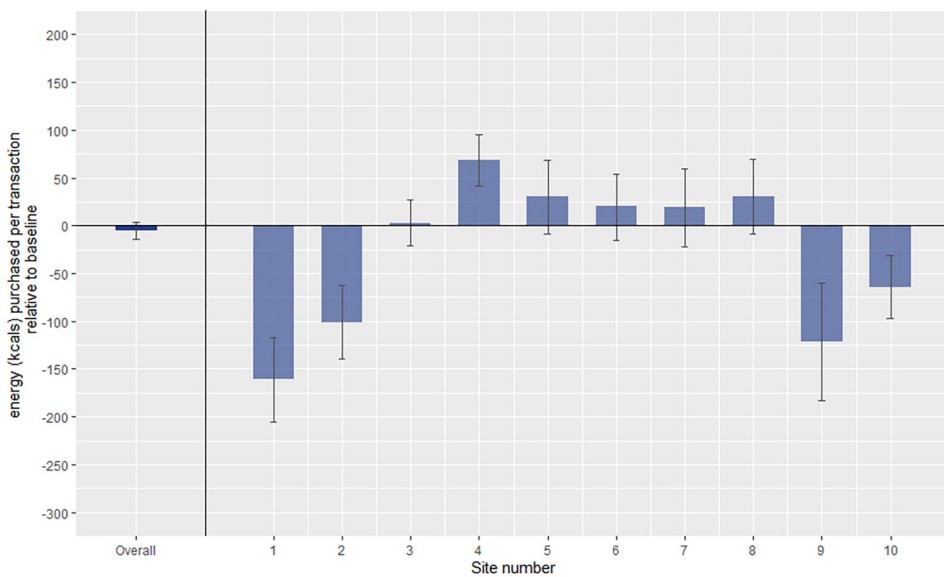

**Fig 4. The effects PACE on energy (kcals) purchased per transaction relative to baseline.** Error bars represent 95% confidence intervals for the overall effects and 99.5% confidence intervals (Bonferroni adjustment) for the by-cafeteria effects.

**Table 3. Full model results for the primary outcome.**

| | Calories M (SE) | 95% CI | p | Pre-intervention mean Daily calories | % Change Post-intervention | 95% CI |
|---|---|---|---|---|---|---|
| Overall model | | | | | | |
| Modelling of the mean (identity link): | | | | | | |
| (Intercept) | 23,495.2 (1,586.6) | (20,385.4, 26,604.9) | <0.001 | | | |
| PACE | −1,934.4 (1,630.8) | (−5,130.8, 1,261.9) | 0.236 | 148,349.5 | −1.30% | (−3.46%, 0.85%) |
| Transactions | 412.7 (6.7) | (399.6, 425.8) | <0.001 | | | |
| Days from start on study period | −112.8 (54.8) | (−220.3, −5.4) | 0.040 | | | |
| Modelling of the variance (log link) | | | | | | |
| (Intercept) | 11.1 (0.1) | | <0.001 | | | |
| Site 2 (Ref = Site 1) | −0.7 (0.1) | | <0.001 | | | |
| Site 3 | −1.1 (0.1) | | <0.001 | | | |
| Site 4 | −0.1 (0.2) | | 0.505 | | | |
| Site 5 | −1.4 (0.1) | | <0.001 | | | |
| Site 6 | −1.2 (0.2) | | <0.001 | | | |
| Site 7 | −0.8 (0.1) | | <0.001 | | | |
| Site 8 | −1.8 (0.1) | | <0.001 | | | |
| Site 9 | −0.9 (0.1) | | <0.001 | | | |
| Site 10 | −0.9 (0.1) | | <0.001 | | | |
| By-site | | | | | | |
| Modelling of the mean (identity link): | | | | | | |
| (Intercept) | 20,986.2 (2,145.1) | (16,781.9, 25,190.4) | <0.001 | | | |
| Site 1 | −65,232 (6,369.7) | (−77,716.4, −52,747.5) | <0.001 | 167,582.0 | −38.93% | (−46.38%, −31.48%) |
| Site 2 | −28,036.2 (3,830.3) | (−35,543.5, −20,528.9) | <0.001 | 114,649.4 | −24.45% | (−31.00%, −17.91%) |
| Site 3 | 1,638.2 (4,849.7) | (−7,867.1, 11,143.4) | 0.736 | 215,735.4 | 0.76% | (−3.65%, 5.17%) |
| Site 4 | 54,458.9 (7,511.3) | (39,737.1, 69,180.7) | <0.001 | 328,017.4 | 16.60% | (12.11%, 21.09%) |
| Site 5 | 6,539.6 (3,050.7) | (560.2, 12,518.9) | 0.032 | 90,466.6 | 7.23% | (0.62%, 13.84%) |
| Site 6 | 6,560.7 (4,138) | (−1,549.7, 14,671) | 0.113 | 138,858.3 | 4.72% | (−1.12%, 10.57%) |
| Site 7 | 7,890 (5,828.1) | (−3,532.9, 19,313) | 0.176 | 165,335.8 | 4.77% | (−2.14%, 11.68%) |
| Site 8 | 4,811.9 (2,249.2) | (403.6, 9,220.2) | 0.033 | 66,040.6 | 7.29% | (0.61%, 13.96%) |
| Site 9 | −24,699 (4,445.2) | (−33,411.5, −15,986.5) | <0.001 | 83,892.8 | −29.44% | (−39.83%, −19.06%) |
| Site 10 | −27,095.3 (4,920.8) | (−36,739.9, −17,450.7) | <0.001 | 175,497.5 | −15.44% | (−20.93%, −9.94%) |
| Transactions | 413 (7.6) | (398.1, 427.9) | <0.001 | | | |
| Days from start on study period | −101.8 (58.2) | (−215.8, 12.2) | 0.081 | | | |
| Modelling of the variance (log link) | | | | | | |
| (Intercept) | 10.4 (0.1) | | <0.001 | | | |
| Site 2 (Ref = Site 1) | −0.4 (0.1) | | 0.002 | | | |
| Site 3 | −0.5 (0.1) | | <0.001 | | | |
| Site 4 | 0.2 (0.1) | | 0.150 | | | |
| Site 5 | −0.8 (0.1) | | <0.001 | | | |
| Site 6 | −0.6 (0.2) | | <0.001 | | | |
| Site 7 | −0.2 (0.1) | | 0.177 | | | |
| Site 8 | −1.2 (0.1) | | <0.001 | | | |
| Site 9 | −0.5 (0.1) | | <0.001 | | | |
| Site 10 | −0.6 (0.1) | | <0.001 | | | |

**Table 4. Unadjusted data (mean [SD]) for daily purchases, revenue, and prices in cafeterias during intervention periods.**

|  | Baseline | PACE |
|---|---|---|
| Energy (kcal) purchased, per cafeteria from intervention categories | 148,350 [87,637] | 163,949 [97,267] |
| Total energy (kcal) purchased, per cafeteria | 149,363 [113,684] | 175,858 [129,821] |
| Number of transactions, per cafeteria | 315 [187] | 365 [204] |
| Revenue (£), per cafeteria | 979 [631] | 1,166 [716] |

A Bayes factor was calculated for the primary outcome: Bayes Factor = 1.03, which suggests there was no evidence that the intervention influences energy purchased.

**Individual cafeteria effects.** There was no evidence that energy purchased was different during the PACE intervention relative to baseline in 5 cafeterias: Cafeteria 3: 1,664 kcals per site per day (99.5% CI −10,963 to 14,290), equivalent to 3 kcals per transaction (99.5% CI −21 to 27); Cafeteria 5: 6,543 kcals per site per day (99.5% CI −2,038 to 15,124), equivalent to 30 kcals per transaction (99.5% CI −9 to 69); Cafeteria 6: 6,571 kcals per site per day (99.5% CI −5,152 to 18,295), equivalent to 20 kcals per transaction (99.5% CI −15 to 54); Cafeteria 7: 7,740 kcals per site per day (99.5% CI −8,682 to 24,162), equivalent to 19 kcals per transaction (99.5% CI −22 to 60); and Cafeteria 8: 4,813 kcals per site per day (99.5% CI −1,496 to 11,123), equivalent to 30 kcals per transaction (99.5% CI −9 to 70).

Energy purchased was significantly lower during the PACE period in 4 cafeterias relative to baseline: Cafeteria 1: −65,204 kcals per site per day (99.5% CI −83,116 to −47,292), equivalent to −161 kcals per transaction (99.5% CI −205 to −117); Cafeteria 2: −27,964 kcals per site per day (99.5% CI −38,733 to −17,197), equivalent to −101 kcals per transaction (99.5% CI −139 to −62); Cafeteria 9: −24,695 kcals per site per day (99.5% CI −37,198 to −12,192), equivalent to -122 kcals per transaction (99.5% CI −183 to −60), and Cafeteria 10: −27,079 kcals per site per day (99.5% CI −41,025 to −13,132), equivalent to −64 kcals per transaction (99.5% CI −97 to −31).

In 1 cafeteria, the PACE intervention significantly increased energy purchased: Cafeteria 4: 54,492 kcals per site per day (99.5% CI 33,251 to 75,733), equivalent to 69 kcals per transaction (99.5% CI 42 to 95).

## Secondary outcomes

**Energy purchased from non-intervention items.** There was no evidence that the PACE labels changed energy purchased from non-intervention items, −5 kcals per site per day (95% CI −513 to 504), $p = 0.986$, equivalent to 0 kcals per transaction (95% CI −1 to 1).

**Total energy purchased (intervention and non-intervention items).** There was no clear evidence that PACE labels changed the total energy purchased from all products, −2,899 kcals per site per day (95% CI −5,810 to 11), $p = 0.051$, equivalent to −8 kcals per transaction (95% CI −16 to 0).

**Revenue.** During the PACE intervention, there was a significant increase in revenue for the cafeterias, £11 per site per day (95% CI 4 to 18), $p = 0.002$, equivalent to £0.03 per transaction (95% CI 0.01 to 0.05).

**Sensitivity analyses.** The 2 sensitivity analyses were consistent with the primary results. First—for the analysis in which we remove some data from the cafeteria that did not provide timely evidence of intervention implementation—there was no evidence that the PACE intervention resulted in an overall change in energy purchased from intervention items, −1,120 kcals per site per day (95% CI −4,392 to 2,153), $p = 0.503$, equivalent to −3 kcals per transaction (95% CI −12 to 6). Second—for the analysis in which the energy estimates were not adjusted

for incorrect button presses—there was no evidence that the PACE intervention resulted in an overall change in energy purchased from intervention items, −1,940 kcals per site per day (95% CI −4,463 to 584), $p$ = 0.132, equivalent to −5 kcals per transaction (95% CI −12 to 2).

**Data checks.**   Results from the main analysis for some of the cafeterias produced larger effect sizes than were predicted. This includes the PACE intervention resulting in −161 fewer kcals purchased per transaction at Cafeteria 1 and 122 fewer kcals purchased per transaction at Cafeteria 9. To provide reassurance about the accuracy of the data collection, we conducted a series of checks to validate the accuracy of these findings, all of which substantiated these findings. First, we checked recording validity at the till sales level to exclude the possibility that a single transaction had erroneously been recorded multiple times (e.g., 50 fish and chips in a single transaction, say). Second, we checked for outliers (defined in the preregistration protocol as 3 units using median absolute deviation) at the aggregate level (daily and weekly) in which energy purchased per cafeteria was examined. Third, we conducted 4 sensitivity analyses on data at the level of the cafeteria: (i) removing outliers; (ii) not adjusting for till button errors (see "Sensitivity analysis" section above); (iii) removing data from the cafeteria that failed to provide evidence of implementation; and (iv) using a model that more simplistically assumes equal variance across cafeterias, which all produced the same result.

## Discussion

In this study, we found no overall evidence that PACE labels changed energy purchased when compared to baseline in 10 worksite cafeterias across England. This conclusion was supported from all 3 indicators of energy purchased: energy purchased from products featuring a PACE label, energy purchased from products not featuring a PACE label, and total energy purchased from all food and drinks. The cafeteria-level analysis showed considerable variation in effects for the primary outcome: Of the 10 cafeterias, there were null results in 5, significant reductions in 4, and a significant increase in 1.

This is the largest naturalistic study to date—to our knowledge—that has evaluated the impact of PACE labels on food and drink purchases with over 250,000 transactions collected via the electronic tills from 10 worksite cafeterias. The pooled results of the current study estimate the effect of PACE labels at −5 kcals per transaction (95% CI −14 kcals to 4 kcals). These results do not overlap with the confidence intervals reported by Daley and colleagues [11]: −80 kcals (95% CI −137 kcals to −24 kcals) nor is the main effect replicated. The current results are therefore not consistent with this review. The current results also appear to be inconsistent with the results of Bleich and colleagues [12] and Viera and colleagues [14]. There are several possible explanations for these apparent differences in outcomes. The Daley review largely included hypothetical, online selection studies and non-naturalistic studies in which participants were recruited in university settings and given menus with PACE labels by researchers, which may not generalise to typical behaviour in restaurants, supermarkets, or cafeterias. Previous labelling studies have suggested that effects in online studies tend to produce larger effect sizes than in lab settings, and lab settings tend to produce larger effect sizes than those observed in naturalistic settings [8,13]. A PACE labelling study in 4 convenience stores [12] found significant effects, but the target of intervention was sugar-sweetened beverages, which only made up a small proportion of the sales in the current study. It remains possible, yet untested, that PACE labels have different effects when applied to different products. A study in 3 cafeterias [14] relied on a subset of the cafeteria's customers sending photos of their meals for 2-week periods every 3 months and therefore the results do not provide a comprehensive account of how the labels influenced overall customer purchases within these cafeterias. In contrast to these studies, the current study applied PACE labels to many categories of food and

drink (hot meals, sandwiches, cold drinks, desserts, etc.) and collected data from every sale for which energy content was available (98%), totalling over 250,000 transactions across 10 cafeterias.

In the current study, we observed a marked difference in the direction and magnitude of the intervention effect in different cafeterias. This variability should sound a note of caution regarding results of trials conducted in small numbers of cafeterias. Contextual factors may have modified the impact of the labels, such as the type of food and drink sold at the cafeterias, the degree of label implementation, or whether the cafeteria is mainly used to eat-in or take away. There may also have been individual differences in those using the different cafeterias, differences that modified their responses to PACE labels. These differences include demographic characteristics including age, gender ethnicity, socioeconomic status, or other factors such as numeracy (for interpreting the values) or weight status. Table A in S1 Additional Data in the supplement shows that the cafeterias varied largely in terms of the food and drink that were regularly sold. For example, the proportion of energy purchased from hot meals ranged from 2% to 27%, breakfasts from 5% to 58%, sandwiches from 4% to 19%, and hot drinks from 0% to 13%. Exploratory correlations suggest that PACE labels show larger effects in cafeterias that sell more discretionary items (e.g., savoury snacks, confectionery). It is possible that PACE labels are more effective at reducing purchasing of discretionary items compared to main meals, which may explain why the effect sizes are larger at certain sites. Although the sample size is too small for any conclusion to be reached from these correlations, it does suggest a testable prediction for future research. Regardless of the underlying explanation for the variation across sites, these possible sources of variation could explain why the results here do not support the conclusions of earlier research. Namely, that the effectiveness of the PACE labels is contingent on contextual factors that differed between the average cafeteria in the current study and the settings used in previous research.

The results of the secondary analyses were consistent with the primary analysis. Namely, there was no clear evidence of an overall effect of the intervention on energy purchased from non-intervention items (i.e., products without PACE labels) or all items (i.e., non-intervention items and intervention items combined). There was evidence that revenue increased during the PACE period relative to the baseline period. As there was no detected effect of the PACE labels on energy purchased, it seems unlikely that this 1.1% increase in revenue was due to addition of the PACE labels, and could be explained by inflation, which increased throughout the Study period by 1.7% as measured using the consumer price index [18] and cannot be easily controlled in a stepped-wedge design. Some studies that have tested different interventions in cafeterias have detected reductions in revenue [15], but there is no evidence that this should be a barrier to implementation of PACE labels.

## Strengths and limitations

The current study was the largest to date (to our knowledge) to implement and test the effectiveness of PACE labels. PACE labels were implemented on the majority of products in cafeterias and outcome variables were measured objectively using data derived from electronic point-of-sale tills. The main limitations of this study were that we were not able to assess consumption of the food and drink purchased, although sales data in cafeterias are normally a good proxy for consumption in these settings as only a small percentage of food is wasted by individuals [19,20]. These sales data were also only described at the transaction level and being able to link transactions to individuals would provide more useful information for inferring causation. Individual identifiers were not possible to acquire due to the companies' desire not to share these data with an external organisation. A potential drawback of all stepped-wedge

designs is that time trends may be partially confounded with the intervention effect estimate; however, as linear time trend was no significant, this is less likely to be an important factor here.

Data collection occurred between April and June 2021 in the UK, during the Coronavirus Disease 2019 (COVID-19) pandemic, and at a time when various government guidelines were in place. This includes stay-at-home restrictions during April and a phased re-opening during May and June. The 2-m social distancing rule was also in place during this time. The cafeterias were still in operation during this time as they were workplace cafeterias within businesses that were exempt from closure; however, it is possible that eating behaviours may have been influenced by the guidelines or wider effects of the pandemic. Although this was not tested, this could include the pandemic affecting food selection or a preference for takeaway foods from cafeterias to avoid crowded spaces. While it is not possible to evaluate the consequences of COVID-19 virus, restrictions, and guidelines on eating behaviours in the current study, the results should be interpreted in light of these contextual factors. These restrictions also meant that fidelity checks needed to be conducted via photographs taken by staff rather than site visits by members of the research team. The accuracy of the intervention fidelity measurements may have been affected by this, particularly if the photographs did not include all products on sale at each site each day, which was not possible to determine without a researcher present.

## Implications for policy and practice

There is no overall evidence from this study that PACE labels would either reduce or increase the amount of energy purchased in worksite cafeterias. However, a meta-analysis of calorie labelling studies suggests that retailers reduced the average energy content of their products by 15 calories after introducing the labels [21]. This suggests that PACE labels—which include calorie content—could have the same effect. It is also possible that providing information on physical activity equivalence may increase activity, although the 1 study that has tested this prediction reported inconclusive results [22]. These possible benefits could support the introduction of PACE or other labelling schemes if accompanied by a formal evaluation to test the full range of impacts. Large-scale rollout would also allow for a formal evaluation of the contextual factors that may influence intervention effectiveness. However, there are concerns about PACE labels that go beyond their effectiveness. There have been claims that labels containing food energy information could exacerbate existing eating disorders; however, there is limited evidence to support this claim to our knowledge [23]. Some evidence comes from an RCT that suggests that a multicomponent intervention which includes PACE labels does not affect emotional eating or uncontrolled eating in a sample where eating disorders were not measured [24]. A pre-post study in university cafeterias that implemented calorie labels found no change in anxiety, body image concerns, or disordered eating among a group at high risk of disordered eating [25]. More evidence is needed before firm conclusions can be reached. A further concern is that PACE labels focus on energy content, rather than other characteristics of the food, they may be ignored by people who are not attempting to lose or maintain their weight.

## Conclusion

The current study provides evidence consistent with PACE labels not influencing food or drink purchasing in 10 cafeterias. Despite an overall null effect, there were significant effects in both directions at individual cafeterias, though mostly towards a reduction due to PACE. Based on these effects and the uncertainties around which contexts could lead to increases or decreases in energy purchased at individual cafeterias, there is insufficient evidence to justify implementing the labels in cafeteria settings. Identifying the characteristics of settings in

which PACE labels could be beneficial is important if these are to be implemented more widely.

## Supporting information

**S1 CONSORT Checklist. Checklist of information to include when reporting a stepped-wedge cluster randomised trial (SW-CRT).**
(DOCX)

**S1 Study Protocol. Effect of physical activity calorie equivalent (PACE) labels on energy purchased in cafeterias: protocol for a stepped-wedge randomised controlled trial.**
(DOCX)

**S1 Additional Data. Supporting tables.**
(DOCX)

## Acknowledgments

We are grateful to everyone involved in running this project including representatives from each company, the catering providers, and the members of staff at every cafeteria.

## Author Contributions

**Conceptualization:** James P. Reynolds, Gareth J. Hollands, Theresa M. Marteau.

**Data curation:** James P. Reynolds, Minna Ventsel, Alice Hobson.

**Formal analysis:** Mark A. Pilling.

**Funding acquisition:** Gareth J. Hollands, Theresa M. Marteau.

**Investigation:** James P. Reynolds, Minna Ventsel, Alice Hobson.

**Methodology:** James P. Reynolds, Mark A. Pilling, Rachel Pechey, Susan A. Jebb, Gareth J. Hollands, Theresa M. Marteau.

**Project administration:** James P. Reynolds, Minna Ventsel, Alice Hobson.

**Supervision:** James P. Reynolds, Rachel Pechey, Susan A. Jebb, Gareth J. Hollands, Theresa M. Marteau.

**Writing – original draft:** James P. Reynolds.

**Writing – review & editing:** James P. Reynolds, Minna Ventsel, Alice Hobson, Mark A. Pilling, Rachel Pechey, Susan A. Jebb, Gareth J. Hollands, Theresa M. Marteau.

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
