## [Editor Report · Decision Letter 0]

28 Feb 2022

Dear Dr Reynolds, 

Thank you for submitting your manuscript entitled "Effect of physical activity calorie equivalent (PACE) labels on energy purchased in cafeterias: a stepped-wedge randomised controlled trial" for consideration by PLOS Medicine.

Your manuscript has now been evaluated by the PLOS Medicine editorial staff and I am writing to let you know that we would like to send your submission out for external assessment.

However, before we can send your manuscript for assessment, we need you to complete your submission by providing the metadata that is required for full assessment. To this end, please login to Editorial Manager where you will find the paper in the 'Submissions Needing Revisions' folder on your homepage. Please click 'Revise Submission' from the Action Links and complete all additional questions in the submission questionnaire.

Please re-submit your manuscript within two working days, i.e. by Mar 02 2022 11:59PM.

Once your full submission is complete, your paper will undergo a series of checks in preparation for assessment. 

Kind regards,

Richard Turner, PhD

rturner@plos.org

---

## [Decision Letter · Decision Letter 1]

5 Apr 2022

Dear Dr. Reynolds,

Thank you very much for submitting your manuscript "Effect of physical activity calorie equivalent (PACE) labels on energy purchased in cafeterias: a stepped-wedge randomised controlled trial" (PMEDICINE-D-22-00643R1) for consideration at PLOS Medicine. 

Your paper was discussed with an academic editor with relevant expertise and sent to independent reviewers, including a statistical reviewer. The reviews are appended at the bottom of this email and any accompanying reviewer attachments can be seen via the link below:

[LINK]

In light of these reviews, we will not be able to accept the manuscript for publication in the journal in its current form, but we would like to invite you to submit a revised version that addresses the reviewers' and editors' comments fully. You will appreciate that we cannot make a decision about publication until we have seen the revised manuscript and your response, and we expect to seek re-review by one or more of the reviewers. 

We hope to receive your revised manuscript by Apr 26 2022 11:59PM. Please email us (plosmedicine@plos.org) if you have any questions or concerns.

Please let me know if you have any questions, and we look forward to receiving your revised manuscript. 

Sincerely,

Richard Turner, PhD

Senior editor, PLOS Medicine

rturner@plos.org

Please adapt the title to begin "Evaluation of ...", crafting the other wording as needed. 

Please quote the study dates in your abstract.

Also, please add some demographic details (e.g., from table 1) to the abstract.

In the abstract and throughout the text, please quote p values alongside 95% CI, where available. 

You report some quantitative details and comparisons with findings of other studies in the second paragraph of the Discussion (main text), and it may be that this element should appear at the end of the Results section. 

Throughout the text, please adapt reference call-outs to the following style: "... 13 different types of cancer [1-3]." and restructure the reference list as appropriate.

Please use the style "... 3 eligibility criteria" throughout, although numbers should be spelt out at the start of sentences. 

In the reference list, please convert all italics into plain text. 

Noting Deery et al, please list no more than 6 author names, followed by "et al.", and remove the ampersands.

Noting Vasiljevic et al, please ensure that all references have full access details. 

Thank you very much for including the CONSORT checklist, and please adapt this so that individual items are referred to by section and paragraph number (not by line or page numbers as these generally change in the event of publication). 

Please also include a completed CONSERVE checklist if appropriate. 

Please include the trial protocol as an attachment, referred to in the Methods section (main text).

Comments from the reviewers:

*** Reviewer #1: 

This is a large trial answering a novel research question of public health relevance. The manuscript is generally well presented and methodologically strong. I therefore am very supportive of this work and believe a revision would strengthen the manuscript. 

Abstract

Inconsistent use of 'ten' vs '10'

Introduction

The introduction is concise and summarises existing PACE labelling literature well. However, it would benefit from revising to better set the scene for why the study was important to conduct. For example, the focus here is on the out of home food sector - most kcal label and PACE literature discussed is tested in that setting and the present study was conducted in the out of home food sector. Why should readers of this journal care about the out of home food sector in relation to medicine/public health? The energy content of foods sold in that setting, how often people eat in that setting and recent policy developments which suggest it is now an area of policy interest (e.g. new kcal labelling law in the UK) spring to mind as being relevant. There is plenty of citeable evidence of relevance - a couple of papers addressing this that we have recently published: https://www.bmj.com/content/363/bmj.k4982 and https://www.bmj.com/content/372/bmj.n40 as an example, but there may be other sources the authors prefer to cite. Similarly, I also think the explanation of why PACE has been considered as an alternative (e.g. some theory and also potentially some reference to public calls for PACE labelling vs. it being considered controversial/questioned by some public health groups) would be informative and build a better case for why this is an important study. A relevant source here - https://www.sciencedirect.com/science/article/pii/S0091743521003820

Method

There are differences between the protocol and analysis protocol on the OSF vs. what is reported in this manuscript. One example is the eligibility criteria, another example is the planned use of Bayesian analyses. There may be more (I haven't looked at in detail to compare and will leave it to the authors to do so). The authors should include a section somewhere on deviations from the original protocol method and planned analyses. 

Page 8, line 9. 'There were largely no energy labels on shelf edge labels or menus'. Likewise on line 6 'most labels'. This is very vague and given that the study is about how changes to labelling affected consumer behaviour, the authors should provide more detailed information (i.e. numbers) and explain how this information was obtained. 

More detail on how fidelity checks were done is needed. Did staff take photos of every single food item pre-intervention each week and then each week during the intervention? The article implicitly reads like this at the moment and I could imagine that over the course of the weeks that the intervention was supposed to be present there would be days when the menu changes and staff forget to do this or over time the labels are less likely to still be where they should be. For example, reading table 2 'yes' vs. 'no' answers seem a simplification of likely compliance. If 1 shelf edge label for 1 food product is photographed, is that evidence of a 'Yes' (I'd assume/hope not), or do x% of food items on a shelf need shelf edge labels to get a 'Yes' (presumably so)?

COVID-19. I'm a bit confused about the relevance of this section. An original plan was referred to, but this isn't in the pre-study protocol or in the clinical trials registry - in these study protocols visits were not planned. I do think a likely weakness of this study is that intervention fidelity could have been more accurately measured (i.e. study visits), hence why I think this section needs to be accurate. 

Analyses

The analysis protocol specifies that total number of transactions would be controlled for in the primary analyses, but based on my reading of the manuscript this doesn't appear to be the case?

An aspect of the analysis I am struggling to understand is that 'to allow for linear trends, the day number of the trial was fitted as a continuous effect' as was 'week of the trial'. Both day of the trial and week of the trial are going to be correlated with/confounded by intervention status, as days and weeks for intervention periods are on average going to be higher than non-intervention periods (i.e. I guess there should be a linear trend between trial day or week and absence vs. presence of the intervention), potentially resulting in multi-collinearity concerns. Is this the case? 

In the full model reported in the supplementary materials (I would recommend the authors either summarise this in text in the main manuscript or provide it as a table) the effect 'Days from start on study period' presumably relates to days as a continuous variable from day 1 to week 1 to day 7 on week 12? (or does it relate to the first day of the intervention to the last day of the intervention for each site) and there is a significant negative trend - over the course of the 12 weeks, fewer kcals are ordered per day. I think this aspect of the data needs to be considered in more detail. The step design makes the implicit assumption that there is no reason effects should change over time until the intervention is implemented (i.e. mean intake before vs. during the step for each site is compared). However, there is a trend over time irrespective of the intervention (it appears). I appreciate the primary model controls for this, but as in my comment above I'm slightly concerned about this approach because the change over time may well be being driven in part by the intervention always being introduced later in time. It would therefore be informative to analyse and report on whether this daily negative trend is a) observed across both the baseline and the PACE period to the same magnitude. If it is then my concerns above are probably not founded. If the trend differs between the two periods then this would be some evidence that the introduction of PACE labelling is changing kcal purchasing. 

As they were planned and the study finds no effect from traditional significance testing, inclusion of the Bayesian analyses would be preferable. 

Page 17, line 1. I think cause and effect terminology should be removed as we don't know that the intervention significantly reduced energy purchased, we just know it's introduction was associated with a decrease. 

Sensitivity analyses. These are well thought out and important to increase confidence in findings, so should be reported in the online supplementary materials for reviewers / readers to interpret. 

Discussion

Page 18, line 25. It is the largest study on PACE labels, but this doesn't necessarily follow that it is the best estimate. There are limitations to this study and as the authors note there are likely to be contextual factors potentially moderating effects of labels (e.g. location etc.) which this study can't account for. We also have no idea how well field studies translate from one country to another or one out of home food sector vs. another (e.g. Bleich et al. looked at PACE labels in convenience stores and therefore their study may be considered a better estimate because a lot of food purchasing happens in such stores), in the same way that it isn't clear how well lab studies translate from lab to real-world, so I would avoid terminology around this being the 'best' study. 

Page 20, line 3. I'm not convinced that visual inspection does show that. Unless the authors can provide some exploratory analyses to support this point then I think this argument needs to be dropped. An alternative explanation is that effects are largest among studies with the largest imbalance between baseline testing period vs. intervention testing period? i.e. in studies in which one of the testing periods was only 4 weeks and therefore any difference between the two testing periods may be driven by shorter testing periods producing a less reliable estimate of what was happening in terms of consumer behaviour. This fits the data on individual site effects pretty much perfectly. Is it a likely explanation? I don't really know and if the authors disagree then it highlights that making any inferences about why some sites appear to have larger effects than others based on just looking at what is similar/different between the sites is shaky ground. 

Page 21, line 3. Typo 'without'

Page 21. Line 18. I think there is better quality evidence to cite here which looks at change over time pre-post implementation of calorie labelling in restaurants (see Zlatevksa meta analysis for some of the studies) as opposed to comparing restaurants with vs. without kcal labelling cross-sectionally. 

*** Reviewer #2: 

My main comment relates to the absence of power calculation. Neither the protocol, SAP nor manuscript includes a power calculation. This raises questions about the a priori ability of this study to detect meaningful reductions in the primary outcome of interest.

Minor/specific questions:

* Please consider removing the results of the subgroup analysis by cafeteria from the abstract

* Please consider including p-values in the abstract. 

* In the methods section, please include a power calculation including all the assumptions made.

* I note that the results of the primary analysis are reported as percentages when the outcome is total energy in kcal (absolute value). According to the analysis plan, available as a preprint, the outcome for the primary analysis is the absolute value (kcal), not a percentage change. If the intention is to take the baseline levels into account due to variations across cafeterias, I would suggest adding the baseline value as a covariate. Please consider reporting absolute values instead or in addition to percentage changes. If presenting percentages, please clarify their meaning in the method/analysis section.

* Please consider including a graph showing the absolute value of the primary outcome (y-axis) per week (x-axis) with one curve for each cafeteria. To indicate when each cafeteria switched to the intervention, one could use different colours or patterns. This would allow the reader to visualise the entire data series.

* Please note that analyses by individual cafeteria were unlikely to be sufficiently powered. When reporting the results (bottom of page 16 + top of page 17), please refrain from saying that the intervention significantly reduced or decreased xxx. It would better to write that "there was a significant difference/reduction/increase between the control and intervention period" without making a causal statement.

* Please add the outcome "total energy purchased" to table 3.

* All outcomes are potentially affected by the number of persons with access to the cafeteria. I note that the analyses do not adjust for changes in exposure (number of persons) and wonder whether this has been / should be considered. It is potentially particularly relevant in the context of the Covid pandemic which may have impacted the number of persons "coming to work". In addition to potential fluctuations in the number of potential customers present, is it also possible that the profile of those present might have changed over time. Are sensitivity analyses possible?

* The analysis model assumes a common time trend for all cafeterias. Given the apparent differences across cafeterias, it might make sense to run a model with varying time trends e.g. by adding a random slope by cafeteria (see Hemming, K., Taljaard, M. & Forbes, A. Analysis of cluster randomised stepped wedge trials with repeated cross-sectional samples. Trials 2017;18:101 https://doi.org/10.1186/s13063-017-1833-7 ). In addition, the effect of time was modelled as a linear time trend and I wonder whether this is a realistic assumption. One might wish to consider not assuming any trend e.g. by considering time/week as a class effect.

- Laurent Billot

*** Reviewer #3: 

This study "Effect of physical activity calorie equivalent (PACE) labels on energy purchased in cafeterias: a stepped-wedge randomized controlled trial" was a stepped-wedge RCT of 10 worksite cafeterias in the UK to test the impact of PACE labels on total energy purchased from intervention labeled items per day compared to baseline. The study took place over 12 weeks between April 6 and June 28, 2021, which coincided with the time that the UK was implementing a phased re-opening of COVID-19-related restrictions. The study found that there was no overall effect of PACE labels on energy purchased but that there was variation of the effect between the different cafeterias. 

This was a well-designed stepped wedge trial that had to be adapted somewhat due to the COVID-19 pandemic. The authors have done an excellent job of explaining the methods, the challenges to implementation, and the strategies to work around some of these challenges. The results are presented in a straightforward manner that is appropriate for demonstrating that there was no effect on the primary outcome.

My comments are therefore more directed at the introduction and discussion. The introduction summarizes the existing literature on PACE labels quite thoroughly. The literature suggests that PACE labels may be effective for reducing energy intake, but there are relatively few studies of PACE labels conducted in real-world environments. However, I think the authors should be more explicit about why it was important for them to conduct their particular study. What are the policy and/or program implications of using PACE labels in a worksite or other setting? In other words, why is it important to do another study of PACE labels, and why in worksites? Despite multiple studies, PACE labels have not had much traction in the real world, and therefore it would be important to understand why the authors think this is an important research direction.

I think the Discussion could include more discussion of the reasons why PACE labels may not be effective in the worksite setting. Much of the Discussion reviews food and cafeteria-level factors that may have influenced the effectiveness of the labels, but there is little discussion about employee-level factors that could have played a role. For example, interpreting these labels requires both numeracy and health literacy skills. Even among highly educated populations, numeracy can vary greatly. Other employee factors that might have influenced how the labels were interpreted or received include baseline physical activity level (e.g., employees with physically active jobs may not pay attention to these labels), weight/BMI, race/ethnicity, SES status, and gender. Finally, the impact of the pandemic on employees' attention and motivation to making healthier food choices could have contributed to a lack of effectiveness; while this was suggested in the limitations, I think this the pandemic was a major factor during the study and could be elaborated in more detail. Finally, a limitation of PACE labels (and calorie labels in general) is that they only focus on energy content, rather than the dietary quality of the food; therefore, people who are not focused on weight loss or on preventing weight gain may not think that PACE labels are relevant to them. The limitations should include a comment about the generalizability of the results for other worksite or retail settings that may serve more socioeconomically and racially/ethnically diverse populations.

*** Reviewer #4: 

The manuscript assessed the effect of PACE labels on purchased on 10 food outlets. The strength of the study is the natural setting in which the experiment was conducted. The manuscript is clearly written. There are important methodological limitations that may affect the quality of the study and challenge the validity of the findings. 

An important limitation, that has remained unaddressed, is the power of the study. With a clustered design, the power of the study to detect the anticipated effect remains unwarranted. It is quite conceivable that the study was underpowered to assess the (possibly small) effects of the labels.

A related concern here, is the potentially different effect that the label has on different food groups. It is unfortunate that the study did not assess effectiveness of the label by different categories of food/beverage items. The authors provide evidence in the manuscript that the labels have previously shown effects on specific categories of food items (e.g., sugar sweetened beverages). The hypothesis that the introduction of the labels would influence purchases of a wider range of food items was possibly too optimistic. Further detail on how the labels have led to changes by categories of food items and/or introduced a shift in purchases between different categories would have been an informative to understand the potential effectiveness of PACE labels on food and beverage purchases.

An additional limitation to the study is information on whether the customers have seen and understood the labels. Some exist surveys or assessment of the use of labels in the shops would provide important additional information to interpret the findings. It remains unclear how the label was developed and tested in the population. A formative study on how the target population understands and interprets the labels would provide the necessary evidence to support the proposed hypothesis tested. Without this information, it is quite possible to assume that the customers simply did not see or understand the labels during their decision-making process in the shops.

The CONSORT checklist is rather of limited use (e.g. no pages numbers are provided) and lacks detail. E.g., authors simply refer to page "Cafeterias, Para 2" for a description of sample size. No critical considerations on power and sample size are found there.

A description of the food offer in the different outlets would be helpful, including a description of other, existing, labels that are used on the food and beverage items.

***

[LINK]

---

## [Decision Letter · Decision Letter 2]

30 Jun 2022

Dear Dr. Reynolds,

Thank you very much for re-submitting your manuscript "Evaluation of physical activity calorie equivalent (PACE) labels’ impact on energy purchased in cafeterias: a stepped-wedge randomised controlled trial" (PMEDICINE-D-22-00643R2) for consideration at PLOS Medicine.

I have discussed the paper with our academic editor and it was also seen again by four reviewers. I am pleased to tell you that, once the remaining editorial and production issues are fully dealt with, we expect to be able to accept the paper for publication in the journal.

[LINK]

Please let me know if you have any questions, and we look forward to receiving the revised manuscript.   

Sincerely,

Richard Turner, PhD

rturner@plos.org

Requests from Editors:

Please adopt a cautious approach to the by-site observations, as requested by the referees.

Early in the abstract, we ask you to adapt the text for greater clarity. The second sentence could be adapted to begin "However, the meta-analysis included only 1 study conducted ..."; with the subsequent sentence becoming "We therefore aimed to estimate ... in the context of a randomized study design.", or similar. 

Again at line 10, we suggest "... was carried out to investigate the effect of PACE labels ...".

Immediately after the abstract, please include a new and accessible 'Author summary' section in non-identical prose. You may find it helpful to consult one or two recent research papers published in PLOS Medicine to get a sense of the preferred style: we generally expect three subsections, each of around three bulleted points, each point consisting of one or two short sentences (usually not repeating quantitative elements already quoted in the abstract). 

If not done already, please refer to the attached study protocol in the Methods section (main text). 

Please rename figure 1 "Participant cafeteria inclusion flowchart" or similar. 

In the Results section (main text), under "Total energy purchased (intervention and non-intervention items)" we ask you to revisit the phrase "... indicating that it is more likely that this intervention decreased ...". To avoid the issue of "more likely than what?", we suggest rewording to "... suggesting that the intervention may have led to a decrease in total energy purchased ...".

We suggest beginning the Discussion section (main text) with "In this study, we found ..." or similar. 

In the subsequent paragraph, please avoid or qualify "This is the largest study to date". 

For reference 1 and any other relevant citations, we suggest spelling out "GBD". 

Noting reference 2, please list no more than 6 author names in each citation throughout the reference list, followed where appropriate by "et al.".

Can a publisher and URL (with accessed date) be added to reference 6?

Immediately after the reference list, "S1 PRISMA checklist" should surely refer to CONSORT. 

Thank you very much for including the completed CONSORT checklist. We note that the entries include "Random", for example, and in the interests of clarity we ask you to expand on these entries (e.g., "Methods: section on randomization"). Converting this to a Word doc may help.

Comments from Reviewers:

*** Reviewer #1: 

The authors have responded to my comments comprehensively, aside from one point.

I am still very unconvinced about their speculative suggestions regarding why some outlets may show an effect and others don't. The exploratory analysis is a series of 9 additional correlations without any adjustment for multiple comparisons using a tiny sample size (n=10). The problem with very small sample sizes is that they are only powered to detect very large and infeasible effect sizes. Munafo et al. have written about this. Others have too - https://onlinelibrary.wiley.com/doi/10.1111/brv.12315

A very conservative adjustment for the large number of analyses conducted (p = 0.01) would render one of the significant effects non-significant and the other significant effect is for an r = -0.86. It is inconceivable that the true size of relationship would be anywhere near that large. As I noted, there would probably be a negative correlation between imbalance between baseline testing period vs. intervention testing period and site effect size. But because it may well be spurious and is just post-hoc guess work based on data / observation fishing from a tiny sample size, I wouldn't ask the authors to run a correlation as I wouldn't have any confidence in the result. Given how tiny the sample size is for these analyses (and therefore how unreliable any analysis would) and how infeasible the result is, I don't think it is questionable/misleading to draw any meaning from it. 

*** Reviewer #2: 

My comments have been adequately adressed. I am not convinced about including the site results, especially the "modelling of the variance" part, in Table 3 but will leave the decision to keep them with the editors.

-Laurent Billot

*** Reviewer #3: 

I commend the authors for being highly responsive to reviewers' comments in their response document and in the manuscript edits.

I have just a few additional comments/concerns about the Abstract:

1) The methods and findings should clarify that the primary and secondary outcomes are changes from baseline in total energy (kcal) purchased from intervention items, non-intervention items, and overall purchases per day. "Change from baseline" is not currently specified in the abstract. 

2) The findings of increased revenue per transaction is statistically significant, but the relevance to policy or intervention implementation is not discussed in the abstract, or for that matter, in the manuscript. At the very least, I can't see a good reason for highlighting this finding in the abstract because it does not seem meaningful, and the authors do not include any discussion about this finding in the discussion section of the manuscript. I think it should be removed from the abstract.

3) I would also consider removing from the abstract the references to the site-specific findings, including the sentence, "Of the 10 cafeterias, there were null results in 5, significant reductions in 4, and a significant reduction in 1" and the conclusion that "There was considerable variation in effects between cafeterias, suggesting potentially important unmeasured moderators." Highlighting the findings of the individual cafeterias seems to counter the authors' main justification for conducting their multi-site PACE study that is described in the Introduction of the manuscript. In the last paragraph of the Introduction, they state, "The limitations of existing naturalistic studies are addressed in 3 ways. . . [including] conducting the study in a larger number of sites to increase the study power and test the generalisability of the main effects to multiple cafeterias," and in the second to last paragraph of the Introduction, they suggest, "Even if an effect is replicated in a real world setting, the high variability in contexts make it hard to predict if an effect reported in one eating retail outlet will generalise to another." Given these strong arguments for the need to include multiple sites to be able to determine if PACE labels are effective and generalizable, I do not think the findings for the individual worksite cafeterias should be featured in the study abstract.

*** Reviewer #4: 

Thank you for addressing my comments and updating the document.

***

[LINK]

---

## [Decision Letter · Decision Letter 3]

27 Sep 2022

Dear Dr Reynolds, 

On behalf of my colleagues and the Academic Editor, Dr. Barry M. Popkin, I am pleased to inform you that we have agreed to publish your manuscript "Evaluation of physical activity calorie equivalent (PACE) labels’ impact on energy purchased in cafeterias: a stepped-wedge randomised controlled trial" (PMEDICINE-D-22-00643R3) in PLOS Medicine.

PRESS

Sincerely, 

Beryne Odeny 

PLOS Medicine